# High Prevalence of Alternative Diagnoses in Children and Adolescents with Suspected Long COVID—A Single Center Cohort Study

**DOI:** 10.3390/v15020579

**Published:** 2023-02-20

**Authors:** Sarah C. Goretzki, Maire Brasseler, Burcin Dogan, Tom Hühne, Daniel Bernard, Anne Schönecker, Mathis Steindor, Andrea Gangfuß, Adela Della Marina, Ursula Felderhoff-Müser, Christian Dohna-Schwake, Nora Bruns

**Affiliations:** 1Department of Pediatrics I, Neonatology, Pediatric Intensive Care, Pediatric Infectiology, Pediatric Neurology, University Duisburg-Essen, Children’s Hospital Essen, 45147 Essen, Germany; 2Center for Translational Neuro- and Behavioral Sciences C-TNBS, University Duisburg-Essen, 45147 Essen, Germany; 3West German Centre for Infectious Diseases (WZI), University Hospital Essen, University Duisburg-Essen, 45147 Essen, Germany; 4Department of Pediatrics III, Pediatric Pulmonology and Sleep Medicine, University Duisburg-Essen, Children’s Hospital Essen, 45147 Essen, Germany

**Keywords:** pediatric Long COVID, post-acute sequelae SARS-CoV-2 infection, post COVID, differential diagnoses, symptom cluster

## Abstract

Background: Long COVID (LC) is a diagnosis that requires exclusion of alternative somatic and mental diseases. The aim of this study was to examine the prevalence of differential diagnoses in suspected pediatric LC patients and assess whether adult LC symptom clusters are applicable to pediatric patients. Materials and Methods: Pediatric presentations at the Pediatric Infectious Diseases Department of the University Hospital Essen (Germany) were assessed retrospectively. The correlation of initial symptoms and final diagnoses (LC versus other diseases or unclarified) was assessed. The sensitivity, specificity, negative and positive predictive values of adult LC symptom clusters were calculated. Results: Of 110 patients, 32 (29%) suffered from LC, 52 (47%) were diagnosed with alternative somatic/mental diseases, and 26 (23%) remained unclarified. Combined neurological and respiratory clusters displayed a sensitivity of 0.97 (95% CI 0.91–1.00) and a negative predictive value of 0.97 (0.92–1.00) for LC. Discussion/Conclusions: The prevalence of alternative somatic and mental diseases in pediatric patients with suspected LC is high. The range of underlying diseases is wide, including chronic and potentially life-threatening conditions. Neurological and respiratory symptom clusters may help to identify patients that are unlikely to be suffering from LC.

## 1. Introduction

Generally, children and adolescents are not severely affected by acute COVID-19, many even remaining asymptomatic [1]. However, varieties of post-infectious conditions have been described, including neurological or cognitive dysfunctions such as fatigue or loss of smell and taste. These post-infectious symptoms, named Long COVID (LC), Post COVID syndrome (PC) or post-acute sequelae SARS-CoV-2 infection (PASC) occur at all ages and can seriously limit daily life [2].

LC is less frequent in children than in adults and the prevalence is higher in girls than in boys [3]. Even though LC was acknowledged early as an emerging public health problem in children and adolescents, a Delphi consensus definition by the WHO was only published at the end of the year 2021 [4]. This may partially explain the wide range of reported prevalence of LC in pediatric patients of between 1 and 13% [5].

While potential explanatory patho-mechanisms for the development of LC have been described, no diagnostic biomarker for LC has been identified [6,7,8,9,10]. The thorough exclusion of other underlying somatic or psychiatric diseases that might cause the presented symptoms is an essential part of the work-up in suspected LC. However, LC may also involve, worsen or lead to other illness, e.g., psychiatric diseases. It may not always be possible to clearly distinguish the cause and consequence of intertwined somatic and psychiatric illness associated with SARS-CoV-2 and LC.

In the context of clinical identification of adult LC patients, symptom clusters associated with these conditions have been reported. However, these clusters frequently overlap and fluctuate over time [11]. The main clusters observed are neurological, respiratory and systemic inflammatory/abdominal [11,12,13,14]. Whether these clusters also apply to the pediatric population is under debate, potentially complicating and delaying the diagnosis of LC. In addition, to our knowledge, there is very limited literature on the nature and frequency of alternative diagnoses found in children and adolescents with suspected LC.

The aim of this study was to determine the rate of suspected LC patients who suffer from other diseases and to investigate whether symptom clusters described in adults also apply to pediatric patients. For this purpose, we analyzed initial symptoms at presentation and the final diagnosis of pediatric patients referred to our Long COVID outpatient department with suspected LC.

## 2. Materials and Methods

Study design and population

This is a retrospective single-center cohort study from a tertiary care center (University Hospital Essen, Germany) including patients under 18 years with a proven history of SARS-CoV-2 infection referred to the Long COVID outpatient department between 1 April 2021 and 30 September 2022 with suspected LC.

Referrals and treatment at the Long COVID outpatient department

Soon after the onset of the pandemic, repeated inquiries for pediatric patients with post-SARS-CoV-2-infection symptoms reached the Pediatric Infectious Disease Department for detailed assessment. Patients were referred from pediatricians, family physicians, primary care physicians and outpatient and inpatient pediatric specialists (Figure 1).

In the early phase of the pandemic, all referral requests were accepted. From January 2021 onwards, patients were triaged after consultation with the referring physicians: acutely ill patients were referred directly to the emergency department at our center. Urgent cases who could not be treated due to limited capacities were referred to emerging Long COVID Outpatient Departments in the surrounding areas. From May 2022 onward, triage included a standardized questionnaire by the German Society of Pediatrics (Deutsche Gesellschaft für Kinder- und Jugendmedizin, DGKJ, Appendix B, Figure 2) which was adjusted by the local physicians [15,16,17,18]. No more than 90 patients per year were treated due to limited capacities (Figure 1).

Verification of SARS-CoV-2 infections and routine diagnostic work-up

All SARS-CoV-2 infections were verified by detection of SARS-CoV-2 IgG and, in vaccinated patients, by the breakdown of SARS-CoV-2 IgG to N and S.

Routine interdisciplinary pediatric subspecialty consultations for all patients comprised specialist of pediatric infectious disease, pediatric pulmonology, pediatric cardiology, pediatric psychiatry or psychosomatics, ophthalmology, and pediatric neurology. Additional consultations were performed if indicated clinically or by exam results. Symptom-oriented instrumental examinations included electroencephalogram, electrocardiogram, echocardiography, abdominal and thyroid ultrasound, chest radiography, lung function assessment including lung clearance index measurement, cranial or spinal magnetic resonance imaging, and nerve conduction velocity, among others.

Self-reported symptoms

Amenorrhea was defined as at least three consecutive missed menstrual cycles. Exercise intolerance was defined as a pathological score in the six minutes walking test (pathological testing equaled: discontinuation due to persisting discomfort, tachycardia, drop in saturation, hypo/hypertension, pathological blood gas analysis). Furthermore a detailed medical history including Post-Exertional Malaise (PEM) Scoring was undertaken in patients with self-reported symptoms of exercise intolerance [19,20].

Data acquisition

The origin of referral, initial symptoms, preexisting conditions, demographics and physical findings were documented on a standardized protocol by the attending physician. Laboratory results (supplementary material, Appendix A), instrumental examinations and findings of interdisciplinary consultations were retrieved from the electronic patient record.

Definition of long COVID

Long-COVID syndrome was defined according to AWMF criteria [21]. NICE criteria for post-COVID condition were additionally assessed [11] (Appendix B, Table A1). Patients who fulfilled AWMF criteria were defined as LC positive if there was no other condition that could explain the symptoms. Patients who did not fulfill AWMF criteria and had another disease diagnosed during the diagnostic process were defined as LC negative. Patients who fulfilled AWMF criteria but had another disease that could explain the symptoms were defined as unclear (Figure 2).

Statistical analyses

Continuous variables are presented as mean and 95% confidence intervals (CI) or standard deviation (SD) if normally or symmetrically distributed and as median and interquartile range (IQR) if skewed. Discrete variables are presented as counts and relative frequencies. Percentages were rounded to integers because of the small total number of observations.

Descriptive statistics were performed for all three subgroups. For further analyses, the group of unclear cases was excluded and the group with clear diagnosis of LC was compared to the group with clear differential diagnoses.

Relative risks and exact 95% CIs were calculated for the registered symptoms to compare the association of symptoms with LC. Symptom clusters that have been reported in adult medicine (neurological, abdominal, respiratory) were calculated and the number of present symptoms for each cluster was counted [12,14]. Receiver operating curves (ROC) and areas under the curve (AUC) with 95% CIs (Wald) were calculated for each cluster using binary logistic regression with LC diagnosis as outcome. Because the discriminatory performance of the abdominal cluster was poor, this cluster was not further investigated.

After frequency analyses of the subgroups for the neurological and respiratory clusters, a data-driven cut off was chosen for ≤2 points vs. >2 points to calculate binary variables if the cluster was “positive” or “negative”. An additional overarching cluster was calculated with positivity of either the neurological or respiratory cluster defining positivity in the combined cluster, and negativity of both the neurological and respiratory cluster defining negativity. Sensitivity, specificity, positive predictive values (PPV), and negative predictive values (NPV) with 95% CIs (Wald) were calculated for all three clusters.SAS Enterprise Guide 8.4 (SAS Institute Inc., Cary, NC, USA) was used to perform statistical analyses and produce figures. 

Ethics approval

The Ethics Committee of the Medical Faculty of the University of Duisburg-Essen approved the study (22-10581-BO). Patient informed consent was waived according to local legislation on retrospective analyses of anonymized data.

## 3. Results

Of 110 patients who presented at our outpatient department, 32 were diagnosed with LC according to AWMF criteria, 21 fulfilled LC criteria but had an additional disease that could explain the symptoms at least partially, and 52 patients did not fulfill LC criteria and were subsequently diagnosed with another somatic or psychiatric disease (Figure 2). The agreement between AWMF and NICE criteria was 100% across all subgroups of patients.

The majority (54%) of patients were referred to our department by pediatricians and originated from across North Rhine-Westphalia and beyond (Figure 3). However, most patients’ residencies had geographic proximity to the hospital.

Mean age at presentation was 12.7 years (152 ± SD 41.33 month) and 41% were male. Patients who had an alternative diagnosis were younger, had less health care system contacts before presentation and displayed lower prevalence of mental illness, depressive symptoms, and suicidal thoughts than LC patients and those who remained unclear (Table 1). Symptoms at presentation included neurological, respiratory, abdominal and unspecific general symptoms as well as psychiatric symptoms (Table 2). The alternative diseases ranged from SARS-CoV-2 associated, autoimmune or neuropediatric diseases to other infections, bronchial asthma or others (Table 3).

16 (15%) patients had been vaccinated against SARS-CoV-2 (Table 1). Except for patients who suffered from residuals after Pediatric Inflammatory Multisystem Syndrome temporarily associated with SARS-CoV-2 (rPIMS-TS), none of the patients had been treated during acute disease or admitted to a hospital (Table 1).

The relative risk of having LC versus another disease was higher when neurological and respiratory symptoms were present, except for cough, whereas in patients with unspecific general symptoms another disease was more likely.

ROC analyses of the calculated symptom clusters showed poor performance of the abdominal cluster (AUC 0.63 (95% CI 0.51–0.74). The neurological cluster had an AUC of 0.84 (0.75–0.93) and the respiratory cluster 0.78 (0.69–0.88). Sensitivity and specificity varied, with higher sensitivity for the neurological cluster (0.75 (95% CI 0.60–0.90)) and higher specificity of the respiratory cluster (0.83 (0.72–0.93)) (Table 4). The negative predictive values of both clusters were higher than the positive predictive values. The combined neurological and respiratory cluster showed an excellent sensitivity of 0.97 (95% CI 0.91–1.00) and a negative predictive value of 0.97 (0.92–1.00) (Table 4).

To facilitate quick screening for potential LC, we developed a simple checklist that relies on the combination of the neurological and respiratory cluster to assess the probability of LC versus a differential diagnosis (Figure 3 and Figure 4). If both clusters are negative, the probability of not having LC is 97%.

## 4. Discussion

This study confirmed a high prevalence of alternative diagnoses in children and adolescents with suspected LC. Of 110 pediatric patients with suspected LC, only one fourth actually fulfilled the definite diagnosis LC and half of the patients were diagnosed with an alternative somatic or psychiatric disease. Another fourth fulfilled LC criteria but had concomitant diseases that could partly explain the symptoms. In contrast to studies conducted in adult patients, none of our patients had a preexisting cardiovascular disease, diabetes or kidney disease [22]. The self-reported duration of the acute SARS-CoV-2 infection did not differ between patients with LC or other diagnoses, contrary to reports from adult studies [23]. As no patient was seriously ill during the acute infection, the influence of disease severity could not be assessed in this study.

Currently, differential diagnoses of LC are mainly discussed in the context of practice guidelines for adults, children and adolescents with LC, but evidence on their prevalence is scarce in adults and absent in children and adolescents [24,25,26]. This study found a large spectrum of differential diagnoses in children and adolescents with suspected LC, ranging from direct sequel by the SARS-CoV-2 virus itself such as residual symptoms after PIMS-TS or myocarditis, to neurological diseases, acute infections, autoimmune diseases and bronchial asthma. In addition, the prevalence of mental disorders was high in our cohort including somatization disorders, depression, anxiety and eating disorders.

At least a quarter of adult patients display concomitant mental illnesses with LC [27,28,29]. A potential explanation for this frequent joint occurrence is the theory that COVID-induced high inflammatory activity of the immune system might promote depression [30]. However, the impact of social isolation, loss of hobbies, interactions with peers, face-to-face teaching, etc. on children and adolescents as a result of preventive measures during the pandemic must not be omitted [31,32,33].

The most important finding of this study is that the high prevalence of alternative diseases in children and adolescents with suspected LC makes extensive somatic and psychiatric diagnostics indispensable to exclude potentially life-threatening acute and chronic diseases. Under no circumstances should LC be diagnosed or treatment initiated without comprehensive diagnostic work, in order to avoid treatment delay of an underlying severe somatic or psychiatric disease.

It is highly relevant that patients with a high likelihood of suffering from an alternative disease are identified quickly in order to initiate adequate diagnostics and therapy. In this cohort, general symptoms, other than chest pain and amenorrhea, were not specific for LC. In contrast, neurological and respiratory symptoms except for cough frequently described in adults with LC were also associated with LC in this pediatric cohort. Regarding abdominal symptoms, rPIMS-TS patients may have introduced bias, which makes the findings of relative risks for abdominal symptoms difficult to interpret. To further investigate the association of clinical symptom patterns and LC, we examined adult symptom clusters regarding their ability to identify or rule out pediatric LC [12,14]. In line with reports on adult patients we found that the neurological and respiratory clusters applied also to pediatric LC patients, whereas unspecific symptoms and the abdominal cluster were not associated with LC [4,34]. The high sensitivity of the respiratory and neurological clusters in identifying LC and, even more importantly, the excellent negative predictive value of the combined clusters may be of practical value in the future to quickly identify patients who are likely to suffer from an alternative disease and require extensive diagnostic work-up. However, use of the checklist does not exempt the attending physician from a thorough medical workup.

There was a group of patients in this study, however, in whom it could not be determined whether residuals of the infection with SARS-CoV-2 caused the complaints or whether it aggravated an underlying disease. This group needs to be further investigated because, contrary to adult patients, there is no evidence as to which underlying diseases predispose children to develop LC [35]. In clinical practice, these patients do not fit into the standard health care system and therefore are in special need of a multi-professional team to rule out differential diagnoses, organize follow-up visits and supportive treatments such as physiotherapies [36].

There are several limitations to this study. The limited capacity of our Long COVID outpatient department (maximum 90 patients per year) potentially resulted in a selection bias towards severe cases. This pre-selected patient population had a high symptom burden and thus a high probability of LC or other diseases. Only five patients had neither LC nor a somatic/psychiatric disease, which does not represent the general pediatric population. Due to the pre-selection, the prevalence of LC may be overestimated by our study. The direct referrals of patients with unambiguous gastroenterological or rheumatological symptoms to the corresponding outpatient departments likely enhanced this phenomenon. In addition, this is a single-center study, which further limits representativeness.

Another important aspect to discuss is the reason why many differential diagnoses were not detected at earlier time points. This may be due to the fact that symptoms of LC are heterogeneous, often vague and broad. For example, Visler et al. describe that more than 200 symptoms can be assigned to the diagnosis of LC []. Furthermore, it is an exclusion diagnosis without established diagnostic biomarkers [37]. Therefore, symptoms may be falsely attributed to LC. In addition, during the pandemic, regular screenings were partially omitted or postponed, potentially delaying the diagnosis of somatic and psychiatric diseases.

## 5. Conclusions

In this study, children and adolescents with suspected LC have a high prevalence of alternative somatic and mental diseases. The range of differential diagnoses includes severe and potentially life-threatening diseases from nearly all pediatric subspecialties. Diagnosis of LC should only be made or “symptomatic” therapy initiated with prior exclusion of other diseases. Using clinical criteria and the adult neurological and respiratory LC clusters, it is possible to identify patients with low probability of LC early and diagnose them in detail.

## Figures and Tables

**Figure 1 viruses-15-00579-f001:**
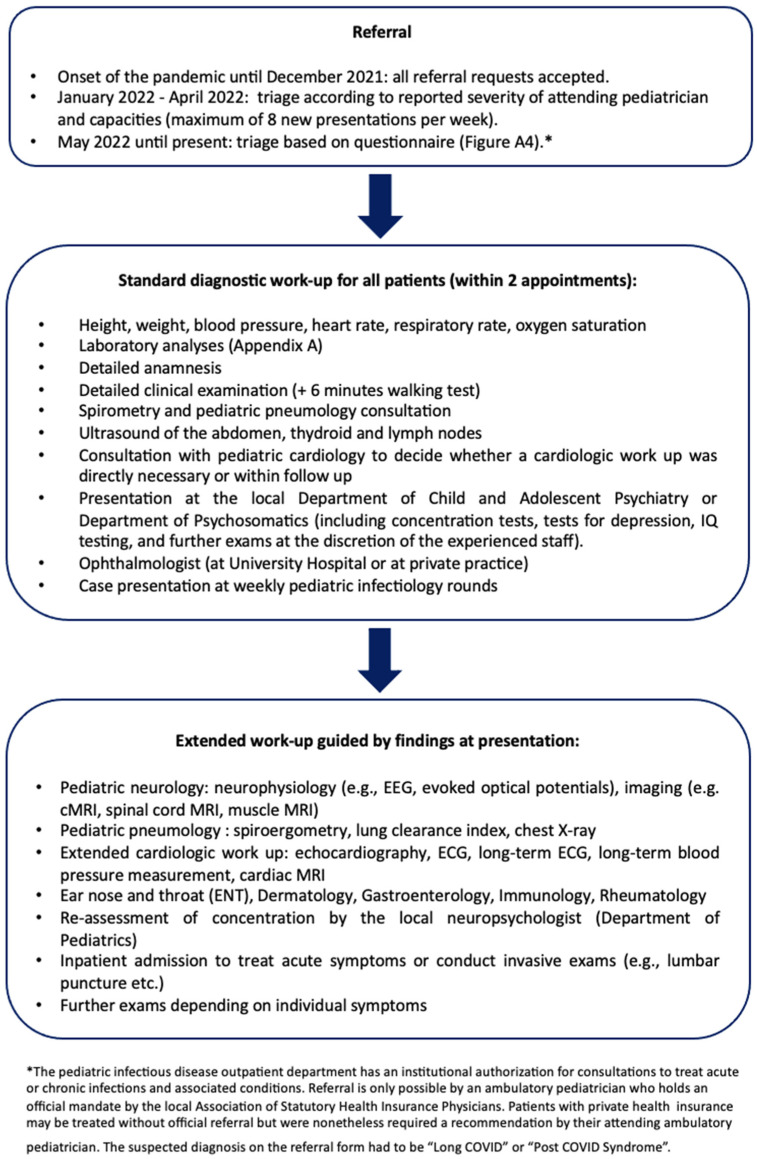
Referral and diagnostic procedure at the Long COVID Outpatient Department at the University Hospital Essen (see Appendix B, Figure A2 for the referral request form).

**Figure 2 viruses-15-00579-f002:**
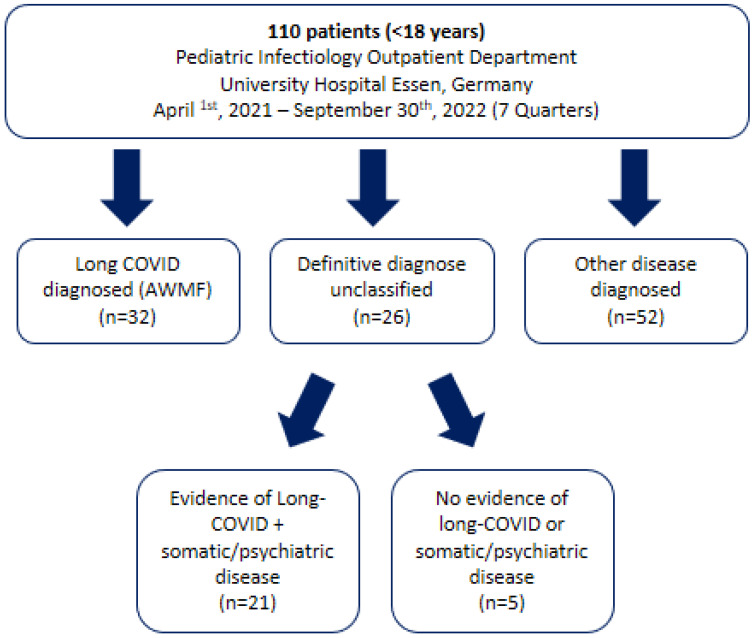
Flow chart of children and adolescents who attended the Long COVID Outpatient Department at the University Hospital Essen between January 2021 and September 2022 with suspected long COVID.

**Figure 3 viruses-15-00579-f003:**
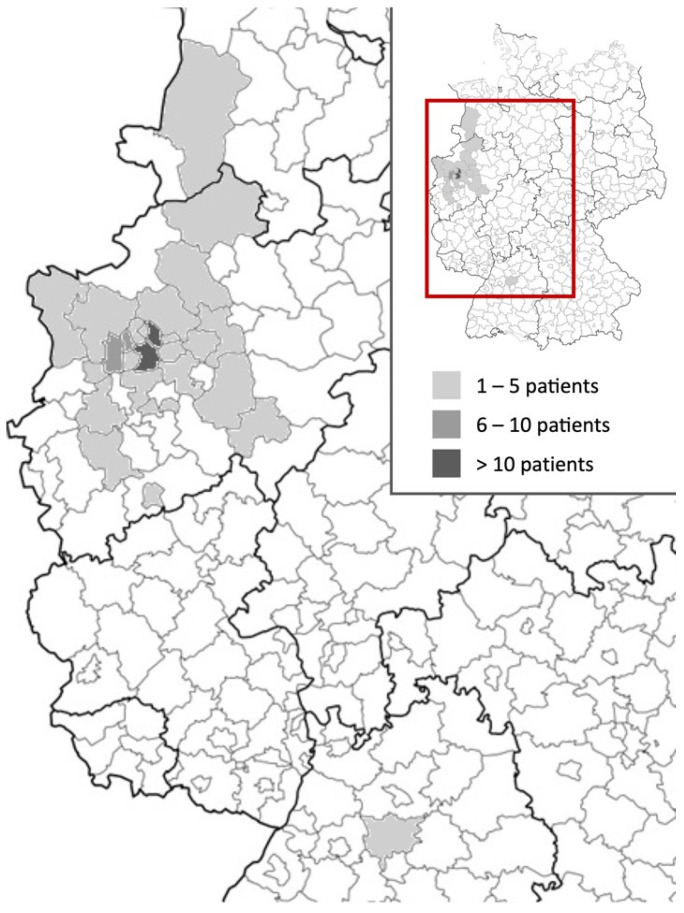
Heat map of referral areas of children and adolescents attended at the Long COVID Outpatient Department of the University Hospital Essen between January 2021 and September 2022 with suspected Long COVID.

**Figure 4 viruses-15-00579-f004:**
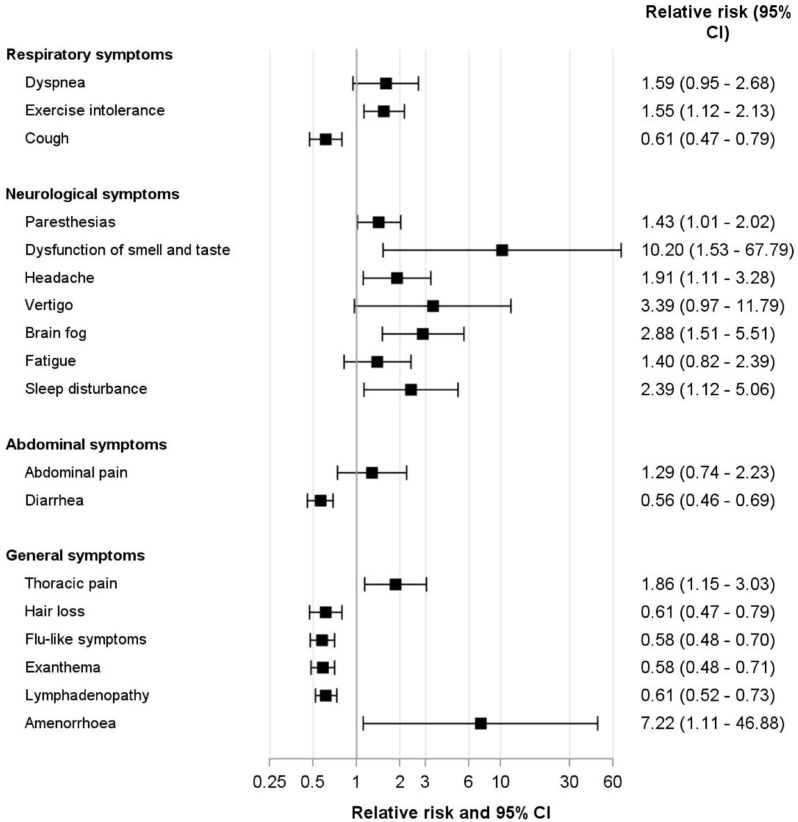
Relative risk of being diagnosed with Long COVID versus an alternative disease in children and adolescents with suspected Long COVID depending on symptoms at presentation.

**Table 1 viruses-15-00579-t001:** Patient characteristics.

		Long COVID	Other Disease	Unclear
		n = 32	n = 52	n = 26
Male		13 (41%)	25 (48%)	13 (46%)
Age (years)	mean (95% CI)	12.1 (10.9–13.4)	9.1 (7.5–10.8)	11.8 (10.1.–13.4)
met AWMF criteria		32 (100%)	0 (0%)	17 (65%)
met NICE criteria		32 (100%)	0 (0%)	17 (65%)
SARS-CoV-2 IgG BAU/mL (breakdown into anti-spike(S) and anti-nucleocapsid(N) SARS-CoV-2-IgG was only carried out for vaccinated patients)	mean ± SD IQR	1194 ± 904 1813 (268–2080)	1016 ± 950 1974 (106–2080)	911 ± 964 2013 (67–2080)
self-reported duration of acute disease (days)	median IQR	3 3 (2–5)	4 5 (3–8)	3 3 (0–4)
Time from infection to first consultation ^6^ (days)	mean ± SD	120 ± 99	120 ± 167	127 ± 130
Number of health care system contacts before presentation	mean (95% CI)	3.2 (2.6–3.8)	2.0 (1.7–2.3)	2.6 (1.5–3.7)
newly diagnosed somatic disease	total	4 (1%)	52 (100%)	16 (62%)
	SARS-CoV-2 associated diseases ^1^	0 (0%)	23 (44%)	0 (0%)
	autoimmune diseases ^2^	0 (0%)	3 (6%)	0 (0%)
	bacterial infections ^3^	0 (0%)	5 (10%)	5 (19%)
	neuropaediatric diseases ^4^	1 (3%)	7 (13%)	3 (12%)
	bronchial asthma	5 (16%)	6 (19%)	8 (31%)
	New-onset anaemia	0 (0%)	0 (0%)	1 (0.4%)
	Others ^5^	0 (0%)	10 (19%)	0 (0%)
newly diagnosed mental disease		14 (44%)	3 (6%)	12 (46%)
psychiatric symptoms	suicidal thoughts	4 (13%)	0 (0%)	3 (12%)
	depression/depressive symptoms	20 (63%)	10 (19%)	12 (46%)
SARS-CoV-2 immunisation	at least 1 vaccination	3 (9%)	6 (12%)	7 (27%)
	≥2 vaccinations	2 (6%)	4 (8%)	6 (23%)
virus waves	Alpha	9 (28%)	16 (31%)	9 (35%)
	Delta	5 (16%)	9 (17%)	11 (42%)
	omicron	18 (56%)	18 (35%)	4 (15%)

SD = standard deviation, AWMF = German Association of the Scientific Medical Societies, NICE = National Institute for Health and Care Excellence; ^1^ SARS-CoV-2 associated diseases = COVID-19, rPIMS-TS, myocarditis due SARS-CoV-2-infection; ^2^ autoimmune diseases = dermatomyositis, rheumatoid arthritis, inflammatory bowel disease; ^3^ bacterial infections = pneumonia, mastoiditis, neuroborreliosis; ^4^ neuro-pediatric diseases = encephalitis, neuritis, myositis, Lambert Eaton/myasthenia, migraine, epilepsy; ^5^ Others = secondary real hypertension, ruptured ovary, immunodeficient, EBV infection, oncological primary disease, HbSS, constipation, astigmatism, parasitosis; ^6^ except 10 unclear infection data (2 of the group “unclear”, 8 of the group “other diagnosis”).

**Table 2 viruses-15-00579-t002:** Symptoms at presentation.

		Symptom Present			Symptom Leading *		
		Long COVID	Other Disease	Unclear	Long COVID	Other Disease	Unclear
		n = 32	n = 52	n = 26	n = 32	n = 52	n = 26
Respiratory symptoms	dyspnoea	12 (38%)	9 (17%)	3 (12%)	8 (15%)	8 (25%)	9 (35%)
	exersice intolerance	25 (78%)	26 (50%)	8 (31%)	12 (23%)	21 (66%)	17 (65%)
	cough	1 (3%)	12 (23%)	0 (0%)	0 (0%)	0 (0%)	3 (12%)
Neurological symptoms	paresthesias	21 (66%)	22 (42%)	1 (4%)	0 (0%)	13 (41%)	14 (54%)
	dysfunction of smell and taste	13 (41%)	1 (2%)	2 (8%)	9 (17%)	1 (3%)	5 (19%)
	headaches	15 (47%)	9 (17%)	2 (8%)	0 (0%)	1 (3%)	10 (39%)
	vertigo	8 (25%)	2 (4%)	1 (4%)	0 (0%)	0 (0%)	8 (31%)
	brain fog	19 (59%)	7 (13%)	0 (0%)	0 (0%)	1 (3%)	12 (46%)
	fatigue	9 (28%)	8 (15%)	0 (0%)	0 (0%)	0 (0%)	10 (39%)
	sleep disturbance	12 (38%)	5 (10%)	0 (0%)	0 (0%)	0 (0%)	7 (27%)
Abdominal symptoms	abdominal pain	7 (22%)	7 (14%)	1 (4%)	0 (0%)	3 (9%)	2 (8%)
	diarrhea	0 (0%)	1 (2%)	0 (0%)	0 (0%)	1 (3%)	0 (0%)
General symptoms	thoracic pain, palpations	17 (53%)	11 (21%)	2 (8%)	3 (6%)	5 (16%)	11 (42%)
	hair loss	1 (3%)	12 (23%)	0 (0%)	0 (0%)	11 (21%)	0 (0%)
	Flue-like symptoms	0 (0%)	8 (15%)	0 (0%)	0 (0%)	4 (13%)	0 (0%)
	exanthema	0 (0%)	7 (13%)	0 (0%)	0 (0%)	1 (3%)	0 (0%)
	lymphadenopathy	0 (0%)	1 (2%)	0 (0%)	0 (0%)	1 (3%)	1 (4%)
	amenorrhoea	9 (28%)	1 (2%)	0 (0%)	0 (0%)	0 (0%)	5 (19%)

* According to patient’s perception.

**Table 3 viruses-15-00579-t003:** Newly encountered diseases in patients with suspected long COVID *^1^.

	Type of Disease	Specific Disease	n (%)
somatic disease	SARS-CoV-2 associated diseases	COVID-19	5 (10%)
		rPIMS-TS	16 (31%)
		myocarditis	2 (4%)
	autoimmune diseases	dermatomyositis	1 (2%)
		rheumatoid arthritis	1 (2%)
		inflammatory bowel disease	1 (2%)
	infections	pneumonia (bacterial)	1 (2%)
		neuroborreliosis	1 (2%)
		EBV infection	2 (4%)
		asymptomatic CMV infection *^2^	1 (2%)
		parasitosis	2 (4%)
		localized bacterial infection	2 (4%)
	neuropediatric diseases	neuritis	1 (2%)
		Lambert Eaton syndrome/myasthenia	5 (10%)
		migraine	1 (1%)
	pulmonary disease	bronchial asthma	6 (12%)
	others	secondary renal hypertension	1 (1%)
		ruptured ovary	1 (1%)
		immunodeficiency	2 (1%)
		oncological/hematological disease	4 (8%)
		constipation	1 (1%)
		astigmatism	1 (1%)
Mental disease		adjustment disorder/somatization disorder	3 (6%)
		depression	1 (1%)

*^1^ Previously known diagnoses are not included, e.g., homozygous sickle cell anemia; *^2^ Considered as an incidental finding rather than the cause of the symptoms after thorough work up. rPIMS-TS = residuals of Pediatric Inflammatory Multisystem Syndrome temporarily-associated with SARS-CoV-2 EBV = Epstein-Barr virus, CMV = cytomegalovirus.

**Table 4 viruses-15-00579-t004:** Sensitivity, specificity, and predictive values to identify patients with LC versus alternative diseases from clinical symptom clusters in suspected pediatric Long COVID.

	Cluster		
	Neurological	Respiratory	Combined *
Patients with positive cluster			
LC (N_total_ = 32)	24 (75%)	21 (66%)	31 (97%)
Non-LC (N_total_ = 52)	10 (19%)	9 (17%)	16 (31%)
Statistical measure			
Sensitivity	0.75 (0.60–0.90)	0.66 (0.49–0.82)	0.97 (0.91–1.00)
Specifity	0.81 (0.70–0.91)	0.83 (0.72–0.93)	0.69 (0.57–0.82)
Positive predictive value	0.71 (0.55–0.86)	0.70 (0.54–0.86)	0.66 (0.52–0.80)
Negative predictive value	0.84 (0.74–0.94)	0.80 (0.69–0.90)	0.97 (0.92–1.00)

* Positive if either the neurological OR the respiratory cluster applies, negative if neither cluster applies.

## Data Availability

Not applicable.

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
