# Peer review of "High Prevalence of Alternative Diagnoses in Children and Adolescents with Suspected Long COVID—A Single Center Cohort Study"

_viruses, 2023, doi:10.3390/v15020579_

Round 1

Reviewer 1 Report

This is a retrospective monocentric study looking at the important question of alternative diagnoses in pediatric patients with suspected long COVID.  The research question is interesting, but the study suffers from several methodical weaknesses, and some of the final diagnoses in those patients raise questions.

-       Introduction

o   L50: There is a Delphi consensus definition for long COVID in children, as well as the AWMF definition and the NICE definition, which the authors use later on. Thus, this sentence should be modified (“there is no common definition”). 

o   L55: “…only be diagnosed after thorough exclusion of underlying somatic or psychiatric disease”. Clearly, LC can also involve, or lead to, psychiatric conditions. Thus, this model of post-COVID symptoms being either somatic, psychiatric or true LC is reductionistic and does not reflect the multifactorial nature of the disease

-       Methods: The weak methods section is a key concern about the manuscript. 

o   How was the history of infection with SARS-CoV-2 proven or was this purely based on history taking? Which serological assays were used to differ between infection and immunization? What was the frequency of a SARS-CoV-2 infection in the patients ultimately receiving another diagnosis?

o   L92: “Routine interdisciplinary pediatric subspecialty consultations comprised pediatric infectious disease specialist, pulmonology, cardiology, psychiatry or psychosomatics, ophthalmology, and neurology”. This is 8 subspecialties. What does routine interdisciplinary consultations mean? For how many patients were all these consultations involved? The routine workup needs to be explained in detail - very importantly including the patient referral and allocation to the LC outpatient department

o   L100: “Self-reported symptoms like amenorrhea or exercise intolerance were objectified”. How can amenorrhea be objectified? Many of the LC symptoms are hard to objectively prove or disprove.

o   The diagnostic algorithm of LC needs to be discussed in detail, given it to be an exclusion diagnosis with many overlaps to other diseases. Just referencing the AWMF and NICE guideline is insufficient (note: the NICE guideline does not have a long COVID, but a post-COVID disease category: https://www.nice.org.uk/guidance/ng188/resources/covid19-rapid-guideline-managing-the-longterm-effects-of-covid19-pdf-51035515742

o   Statistics:

§  L124: “all presented variables are normally distributed”. This sounds very surprising, are the authors confident that all variables are normally distributed?

§  The statistics need to be explained in more detail. How were CIs calculated?

-       Results

o   The LC patient cohort should be better described: Which diagnostic algorithm was used? Which serology tests? Duration of symptoms, onset with respect to SARS-CoV-2 infection?Severity? Psychological and cognitive tests? …

o   The result tables should report statistics (confidence intervals, p-values)

o   Strongest critique point of the manuscript: I have major issues with the final diagnoses that were eventually made in the patients referred as suspected LC. The most frequent alternative diagnosis is PIMS-TS (16/52 patients, 31%). PIMS-TS is an acute-onset life-threatening syndrome of severe hyperinflammation, with at least 2/3 of patients requiring intensive care treatment, the majority including inotropic support. Why and how were these patients referred with suspected LC, a condition that requires symptom duration of at least 12 weeks? Also other final diagnoses are unclear to me: what is transitory adjustment disorder? CMV infection in children is rarely symptomatic, if not of congenital/perinatal origin, or in the context of severe immune dysfunction. 

o   The final diagnoses raise the question: Who made the medical decision to refer a given patient to the “LC outpatient department” of the center? Were proposed referrals to this department screened by a physician based on written patient history prior to the visit? 

o   Table 3: the legend is unclear to me, or at least confusing due to the page break in the pdf. Column 1

o   Table 2: differentiating between the presence of a symptom and it being chief complaint seems attractive; however, from a clinical standpoint one would imagine that many patients with a combination of symptoms can not clearly define one leading symptom.

-       Figure 2 is not discussed

-        Screening survey to identify patients with high probability of an alternative diagnose in suspected pediatric Long COVID: 

o   the item fatigue is listed both in the respiratory and neurological cluster, this should not be the case

o   how many of the 32 LC patients did show the symptom combination to fulfill the respiratory, neurological or both clusters?

-       Generally: language weaknesses, for example

o   L19 “…after exclusion of alternative somatic and mental (XXXmissingXXX).”

o   L65… rate of suspected LC patients who suffer from other disease

o   L300 endocrinologgy

o   L309 standard diagnostics test

o   Figure 1 “definitive diagnose”

o   

Author Response

Dear Reviewer,

thank you for giving us the opportunity to improve our manuscript entitled "High prevalence of alternative diagnoses in children and adolescents with suspected Long COVID - a single center cohort study". We appreciate your thorough review and helpful comments that will certainly improve our manuscript.

Please find below our revisions addressed in point-by-point fashion. Changes to the original manuscript were highlighted in yellow. 

We hope that our revisions meet your expectations.

Yours sincerely,

Sarah Goretzki

- Introduction

o   L50: There is a Delphi consensus definition for long COVID in children, as well as the AWMF definition and the NICE definition, which the authors use later on. Thus, this sentence should be modified (“there is no common definition”). 

  • We modified the sentence and added a new figure (Table. A1) to show the used definitions in more detail: “Even though LC was acknowledged early as an emerging public health problem in children and adolescents, a Delphi consensus definition by the WHO was only published at the end of the year 2021 [30].”

o   L55: “…only be diagnosed after thorough exclusion of underlying somatic or psychiatric disease”. Clearly, LC can also involve, or lead to, psychiatric conditions. Thus, this model of post-COVID symptoms being either somatic, psychiatric or true LC is reductionist and does not reflect the multi-factorial nature of the disease

  • We added this aspect to the introduction and included some of your wording because it stated very well LC as a multidimensional disease: “The thorough exclusion of other underlying somatic or psychiatric diseases, that might cause the presented symptoms, is an essential part of the work-up in suspected LC. However, LC may also involve, worsen or lead to other illness, e.g., psychiatric diseases. It may not always be possible to clearly distinguish the cause and consequence of intertwined somatic and psychiatric illness associated with SARS-CoV-2 and LC.”

-       Methods: The weak methods section is a key concern about the manuscript.

o   How was the history of infection with SARS-CoV-2 proven or was this purely based on history taking? Which serological assays were used to differ between infection and immunization? What was the frequency of a SARS-CoV-2 infection in the patients ultimately receiving another diagnosis?

  • All infections were confirmed by detection of SARS-CoV-2 IgG and, in vaccinated patients by break down into N and S. We added this information to the methods section: All SARS-CoV-2 infections were verified by detection of SARS-CoV-2 IgG and, in vaccinated patients, by the breakdown of SARS-CoV-2 IgG to N and S.“
  • Only patients with confirmed infections we eligible. We added this to the methods section, as well: This is a retrospective single-center cohort study from a tertiary care center (University Hospital Essen, Germany) including patients < 18 years with proven history of SARS-CoV-2 infection referred to the Long COVID outpatient department between April 1st,, 2021 and September 30th, 2022 with suspected LC.“

o   L92: “Routine interdisciplinary pediatric subspecialty consultations comprised pediatric infectious disease specialist, pulmonology, cardiology, psychiatry or psychosomatics, ophthalmology, and neurology”. This is 8 subspecialties. What does routine interdisciplinary consultations mean? For how many patients were all these consultations involved? The routine workup needs to be explained in detail - very importantly including the patient referral and allocation to the LC outpatient department

  • All patients received the routine work-up and some received additional work-up. We added Figures 1 and A2 to specify the work flow from referral to diagnosis.

o   L100: “Self-reported symptoms like amenorrhea or exercise intolerance were objectified”. How can amenorrhea be objectified? Many of the LC symptoms are hard to objectively prove or disprove.

  • That is true. We revised the paragraph for more clarity and added the definition for exercise intolerance: “Amenorrhea was defined as at least three missed menstrual cycles. Exercise intolerance was defined as a pathological score in the six minutes walking test (pathological testing equaled: Discontinuation due to persisting discomfort, tachycardia, drop in saturation, hypo/hypertension, pathological blood gas analysis). Furthermore a detailed medical history including Post-Exertional Malaise (PEM) Scoring was taken [18].“  

o   The diagnostic algorithm of LC needs to be discussed in detail, given it to be an exclusion diagnosis with many overlaps to other diseases. Just referencing the AWMF and NICE guideline is insufficient (note: the NICE guideline does not have a long COVID, but a post-COVID disease category: https://www.nice.org.uk/guidance/ng188/resources/covid19-rapid-guideline-managing-the-longterm-effects-of-covid19-pdf-51035515742

  • We hope we could show the algorithm by showing the diagnostic work-up more clearly and adding a list of used definitions. To acknowledge the remaining uncertainty upon diagnosing or excluding LC, we included the category of “unclear” patients in this study, which comprised approximately one fourth of all patients.

  • Statistics:

  • L124: “all presented variables are normally distributed”. This sounds very surprising, are the authors confident that all variables are normally distributed?

  • In fact, there were only two continuous variables included in the previous version of our manuscript, which were symmetrically distributed. For the revision, we added more continuous variables, some of which are highly skewed. Therefore, we revised the sentence accordingly: “Continuous variables are presented as mean and 95 % confidence intervals (CI) or standard deviation (SD) if normally or symmetrically distributed and as median and interquartile range (IQR) if skewed.“´

  • The statistics need to be explained in more detail. How were CIs calculated?

  • For table 1, we calculated CIs based on a normal distribution, exact CIs for relative risks, and Wald CIs for areas under the curve and Sensitivity-Specifity calculations. We added this information to the methods section accordingly at the location where the calculations are described.

-       Results

o   The LC patient cohort should be better described: Which diagnostic algorithm was used? Which serology tests? Duration of symptoms, onset with respect to SARS-CoV-2 infection?Severity? Psychological and cognitive tests? …

  • We included a flow-chart of our testing, as well as a list of blood work to describe this in more detail. Furthermore we added the duration and severity of the primary infection and stated it in the manuscript. Furthermore we stated the duration between primary infection and first visit in our outpatient department. Psychological testing was decided individually for each patient by the child and adolescent psychiatry and psychosomatics departments. Since there were also very different complaints here.

o   The result tables should report statistics (confidence intervals, p-values)

  • Table 1 contains CIs for age and the number of healthcare contacts. Accounting for the fact that the frequencies we report are very low, it does not seem helpful to calculate CIs for frequencies.
  • For all other relevant statistics (AUC, relative risk, sensitivity, specifity, negative and positive predictive values) CIs are provided. P-values collapse effect strength, precision of the estimation and case number into one single number that is considered statistically significant, thereby drawing the attention away from clinical relevance. Further, they require a prior hypothesis that is to be tested. As we did not have a hypothesis and CIs provide a more appropriate measure for the reader to estimate the effect and the precision of measurement, we prefer to stick with CIs. However, statistical significance can be extrapolated from non-overlapping confidence intervals.

o   Strongest critique point of the manuscript: I have major issues with the final diagnoses that were eventually made in the patients referred as suspected LC. The most frequent alternative diagnosis is PIMS-TS (16/52 patients, 31%). PIMS-TS is an acute-onset life-threatening syndrome of severe hyperinflammation, with at least 2/3 of patients requiring intensive care treatment, the majority including inotropic support. Why and how were these patients referred with suspected LC, a condition that requires symptom duration of at least 12 weeks? Also other final diagnoses are unclear to me: what is transitory adjustment disorder? CMV infection in children is rarely symptomatic, if not of congenital/perinatal origin, or in the context of severe immune dysfunction.

  • Thank you for pointing out this weakness. Indeed, it was not clear enough that PIMS-TS patients actually suffered from residual symptoms after PIMS-TS rather than acute illness. To clarify this, we introduced the term residual PIMS-TS (rPIMS-TS) and a corresponding explanation within the manuscript
  • An adjustment disorder is a transitory state of emotional stress in response to an external stress or (coded in the F43 chapter of the ICD-10-CM system along with reaction to severe stress and post-traumatic stress disorders)
  • The CMV infection was an incidental finding and unlikely related to the symptoms at presentation. We added the information that it was asymptomatic.

o   The final diagnoses raise the question: Who made the medical decision to refer a given patient to the “LC outpatient department” of the center? Were proposed referrals to this department screened by a physician based on written patient history prior to the visit?

  • The triage and diagnostic work flow are shown in figures Figure 1 and A2.

o   Table 3: the legend is unclear to me, or at least confusing due to the page break in the pdf. Column 1

  • We modified the legend for more clarity. It now reads Newly encountered diseases in patients with suspected long COVID*

*Previously known diagnoses are not included, e.g., homozygote sickle cell anaemia

o   Table 2: differentiating between the presence of a symptom and it being chief complaint seems attractive; however, from a clinical standpoint one would imagine that many patients with a combination of symptoms can not clearly define one leading symptom.

  • That is true, and in fact it is the reason why all calculations were performed based on the presence of a symptom and not the self-reported leading symptom. However, we did not want to exclude this information, because there may be discrepancy between the self-considered main symptom and the most serious symptoms from a medical point of view.

  • Figure 2 is not discussed.
  • We added a paragraph in the discussion on Figure 2, which is now figure 4: It is highly relevant that patients with a high likelihood of suffering from an alternative disease are identified quickly in order to initiate adequate diagnostics and therapy. In this cohort , general symptoms, other than chest pain and amenorrhea, were not specific for LC. In contrast, neurological and respiratory symptoms except for cough frequently described in adults with LC were associated with LC in this pediatric cohort, as well. Regarding abdominal symptoms, rPIMS-TS patients may have introduced bias, which makes the findings of relative risks on abdominal symptoms difficult to interpret. To further investigate the association of clinical symptom patterns and LC, we examined adult symptom clusters regarding their ability to identify or rule out pediatric LC [11, 13].”

-        Screening survey to identify patients with high probability of an alternative diagnose in suspected pediatric Long COVID:

o   the item fatigue is listed both in the respiratory and neurological cluster, this should not be the case

  • Fatigue is part of both adult clusters which we assumed without adaptation.

o   how many of the 32 LC patients did show the symptom combination to fulfil the respiratory, neurological or both clusters?

  • We added the number of patients to table 4.

-       Generally: language weaknesses, for example

  • We apologize for the linguistic weaknesses and hope that we were able to identify and correct all errors.

Reviewer 2 Report

Please fine the points below.

1. Abstract line 1: somatic and mental with no noun phrase at the end of the line does not seem meaningful.

2. Lines 74-76: “Retrospective single-center cohort study from a tertiary care center (University  Hospital Essen, Germany) including patients < 18 years referred to the Long COVID outpatient department between April 1st, , 2021 and September 30th  , 2022 with suspected  LC” needs to be completed with a verb phrase.

3. In the methods section the definition of LC must be clarified in terms of the duration of the related symptom, like more than 12 weeks which is normally considered as long-COVID.

4. Is the probable correlation between comorbidities and risk of LC assessed? The present data has divided the symptoms into categories of LC and other diseases. Nevertheless, the association between having an underlying disease and risk of developing LC is not clear. This could be of high value.

5. More studies could be included from other countries to be compared in discussion section (PMID: 35690233, PMID: 35594336).

Author Response

Dear Reviewer,

thank you for giving us the opportunity to improve our manuscript entitled "High prevalence of alternative diagnoses in children and adolescents with suspected Long COVID - a single center cohort study". We appreciate your thorough review and helpful comments that will certainly improve our manuscript.

Please find below our revisions addressed in point-by-point fashion. Changes to the original manuscript were highlighted in yellow. 

We hope that our revisions meet your expectations.

Yours sincerely,

Sarah Goretzki

  • Abstract line 1: somatic and mental with no noun phrase at the end of the line does not seem meaningful.
    • We added the word disease to the end of the sentence: Long COVID (LC) is a diagnosis that requires exclusion of alternative somatic and mental diseases.”

  • Lines 74-76: “Retrospective single-center cohort study from a tertiary care center (University  Hospital Essen, Germany) including patients < 18 years referred to the Long COVID outpatient department between April 1st, , 2021 and September 30th  , 2022 with suspected  LC” needs to be completed with a verb phrase.

  • We rephrased the sentence: This is a retrospective single-center cohort study from a tertiary care center (University Hospital Essen, Germany) including patients under 18 years with proven history of SARS-CoV-2 infection referred to the Long COVID outpatient department between April 1st,, 2021 and September 30th, 2022 with suspected LC.”

  • In the methods section the definition of LC must be clarified in terms of the duration of the related symptom, like more than 12 weeks which is normally considered as long-COVID.
    • We added Table A1 to clearly present the LC definitions available and the one we used.

  • Is the probable correlation between comorbidities and risk of LC assessed? The present data has divided the symptoms into categories of LC and other diseases. Nevertheless, the association between having an underlying disease and risk of developing LC is not clear. This could be of high value.
    • Also the comments that comorbidities can have an influence is very important and we had not addressed it enough so far, so we added: „In contrast to studies conducted in adult patients, none of our patients had a preexisting cardiovascular disease, diabetes or kidney disease [33].“ and „There was a group of patients in this study, however, in whom it could not be determined whether residuals of the infection with SARS-CoV-2 caused the complaints or whether it aggravated an underlying disease. This group needs to be further investigated because, contrary to adult patients, there is no evidence which underlying diseases predispose children to develop LC [32].

  • More studies could be included from other countries to be compared in discussion section (PMID: 35690233, PMID: 35594336).
    • We thank you for these good examples of literature and added these and some more literature to the discussion (references 30-35).
